# Fertility in Celiac Disease: The Impact of Gluten on Male and Female Reproductive Health

**DOI:** 10.3390/nu17091575

**Published:** 2025-05-03

**Authors:** Herbert Wieser, Carolina Ciacci, Carlo Soldaini, Carolina Gizzi, Lucienne Pellegrini, Antonella Santonicola

**Affiliations:** 1Hamburg School of Food Science, Institute of Food Chemistry, University of Hamburg, 20146 Hamburg, Germany; h.wieser2@gmx.de; 2Gastrointestinal Unit, Department of Medicine, Surgery and Dentistry “Scuola Medica Salernitana”, University of Salerno, 84131 Salerno, Italy; carlo.soldaini@outlook.com (C.S.); carolinagizzi@libero.it (C.G.); lucienne.pellegrini@sangiovannieruggi.it (L.P.); asantonicola@unisa.it (A.S.)

**Keywords:** celiac disease, gluten, gluten-free diet, infertility

## Abstract

Celiac disease (CeD) is a chronic immune-mediated disorder of the small intestine triggered by the ingestion of dietary gluten. This narrative review aims to summarize and critically evaluate the recent literature on the association between CeD and infertility, with an emphasis on identifying patterns and inconsistencies. Previous studies have reported conflicting findings: while some demonstrate a higher prevalence of unexplained infertility in patients with CeD, others do not support this association. Overall, untreated CeD may be a contributing factor to infertility, especially unexplained cases, and a gluten-free diet (GFD) might improve fertility outcomes. However, the general prevalence of infertility in CeD patients does not appear to exceed that of the general population. This review includes evidence on both male and female infertility and examines possible pathophysiological mechanisms, including nutritional deficiencies, immune-mediated effects, and sexual dysfunction. Further high-quality prospective studies are needed to determine the true impact of CeD on reproductive health and to inform screening guidelines.

## 1. Introduction

Celiac disease (CeD), also known as gluten-sensitive enteropathy, is a chronic immune-mediated enteropathy that occurs in genetically predisposed individuals upon ingestion of gluten, which is found in wheat, rye, barley, and oat products [1]. Pathologically, CeD is characterized by damage to the small-intestinal mucosa, including villous atrophy, crypt hyperplasia, and increased lymphocyte infiltration. The global prevalence of diagnosed CeD is approximately 1%; however, many cases remain undiagnosed, which increases the risk of long-term complications. CeD can affect individuals of any age, both children and adults, with women being at least twice as likely to be affected as men.

Diagnosing CeD is challenging and involves a multi-step process that includes symptom evaluation, serological testing, duodenal biopsy, response to a gluten-free diet (GFD), and, optionally, genetic testing. Due to this complexity, delays in diagnosis, which can sometimes exceed ten years, are common. The only effective treatment currently available is a strict, lifelong GFD, which typically leads to relief of clinical symptoms and healing of small-intestinal damage.

The clinical presentation of CeD is highly variable, ranging from asymptomatic cases to severe symptoms that can be categorized into gastrointestinal and extra-gastrointestinal manifestations. Common gastrointestinal symptoms include chronic diarrhea, abdominal pain, bloating, flatulence, and vomiting [2], which often prompt physicians to initiate diagnostic procedures for CeD. In contrast, extra-gastrointestinal manifestations—such as nutritional deficiencies, bone-related disorders, dental abnormalities, skin conditions, and neurological or psychiatric disorders [3]—are more difficult to detect. Many of these cases remain undiagnosed, particularly in the absence of gastrointestinal symptoms.

Among the extra-intestinal manifestations of CeD, various adverse reproductive outcomes have been reported, including infertility, spontaneous abortion, stillbirth, preterm delivery, and low birth weight [4,5,6,7,8]. Even endocrine conditions like hypogonadotropic hypogonadism, which can lead to infertility, are linked to celiac disease [9,10,11,12].

According to the World Health Organization (WHO), overall infertility is defined as the “failure to achieve a clinical pregnancy after 12 months or more of regular unprotected sexual intercourse”. Primary infertility refers to the inability to conceive in couples who have never achieved pregnancy, while secondary infertility refers to difficulties in conceiving after previously achieving pregnancy. Unexplained infertility is diagnosed when no apparent cause of infertility can be identified, in both partners, after comprehensive evaluation, but there is no single universally accepted definition of this condition. Unexplained infertility is a diagnosis of exclusion for couples who do not fit the criteria for diagnosis of male factor infertility, oligo/anovulatory infertility or anatomical concerns such as blocked fallopian tubes, endometriosis, uterine cavity defects, or cervical/vaginal obstruction. Primary research studies that recruit couples with unexplained infertility use widely varying inclusion criteria and often do not define their criteria at all [13].

Infertility can stem from female factors, including ovarian, tubal, or endometrial dysfunction, or male factors, such as erectile dysfunction, hypogonadism, or reduced sperm motility. Known causes of infertility include anatomical abnormalities, endocrine disorders, endometriosis, and infections such as Chlamydia trachomatis and tuberculosis. When no identifiable cause is found, an underlying condition such as CeD may be responsible [14,15].

Since the 1970s, numerous studies have explored the association between CeD and infertility. Wang et al. [16] described at least one autoimmune disease in 25% of women with primary ovarian insufficiency with an increased relative risk for CeD of 7.58 [3.47, 14.39]; *p* = 4.47 × 10^−6^). However, the findings remain inconsistent, leaving this issue open to debate.

The aim of this review is to summarize recent studies on the relationship between CeD and infertility to raise awareness among physicians about the potential presence of CeD in patients experiencing reproductive difficulties. Numerous studies have explored the association between CeD and infertility; however, the findings remain inconsistent. Previous reviews have focused predominantly on female infertility or have presented limited updates on newer evidence [17]. This review differs by including studies up to July 2024 and analyzing findings across genders and couples, thereby providing a broader perspective. It also aims to highlight inconsistencies and identify knowledge gaps, offering a critical narrative rather than a meta-analytic synthesis.

## 2. Materials and Methods

### PubMed Search Strategy and Selection Criteria

This is a narrative review. A search of the PubMed database was conducted for articles published in English between 2011 and March 2025. The keywords used were “c(o)eliac disease” combined with “infertility” and “fertility”. Articles without abstracts, such as case reports, commentaries, conference papers, and letters, were excluded. The initial search identified 84 articles, of which 29 were selected for this review. Additionally, 13 papers were identified through cross-referencing the retrieved articles. We conducted a similar search on Scopus, initially identifying 73 articles in English on fertility and celiac disease; among them, we selected 36 publications: a total of 23 were already included in the PubMed search, while 13 were not. Therefore, we added data from these 13 studies to our review (Figure 1).

## 3. Results

In 1970, Morris and colleagues were the first to describe the relationship between untreated CeD and female infertility, as well as the impact of a GFD. They reported three cases of infertile women with untreated CeD who successfully conceived after adopting a GFD [18]. Since then, numerous clinical studies have explored the association between CeD and infertility, but their findings have been inconsistent [4,6,7,8,19].

Several factors may explain the variability in study results, including differences in how CeD (seropositive vs. biopsy-confirmed) and infertility (WHO definition vs. referral to infertility clinics) were defined, as well as variations in the sample sizes and demographic compositions of study populations and control groups. The limited number of participants in most studies (typically fewer than 200) reduces the statistical power of the findings. Additionally, inconsistent data interpretation and a lack of differentiation between overall infertility and unexplained infertility have contributed to the conflicting results. In cases of explained infertility, the underlying causes were rarely detailed.

Screening for undiagnosed CeD among infertile individuals was primarily conducted using serological tests for antibodies against transglutaminase 2 (TGA), endomysium (EMA), and/or gliadin (AGA). However, confirmation of CeD through duodenal biopsy was infrequent.

An overview of relevant clinical studies, published from 2011 to March 2025, is presented in the following subsections. They include the association between CeD and infertility of women, men, and couples and discuss possible pathophysiological mechanisms and the role of a GFD. TGA measurement as a screening method for CeD was the only suitable basis for the comparison of results.

### 3.1. Infertility in Women

To date, the precise risk estimate of infertility in women with CeD remains unclear due to methodological variations and heterogeneity across studies. Furthermore, the impact of a GFD on infertility has not been fully clarified. Findings published before 2011 were reviewed by Lasa et al. [20] and Tersigni et al. [21]. These reviews indicated that undiagnosed CeD appears to be a significant risk factor for infertility in women, and screening for CeD should be considered in cases of unexplained infertility.

Subsequent investigations into the association between CeD and infertility in women have also produced contradictory results.

Table 1 summarizes the prevalence of TGA-positivity among women with overall infertility and unexplained infertility compared to fertile control women.

#### 3.1.1. Positive Associations

Several studies investigating the prevalence of undiagnosed CeD in women with fertility problems have identified a clear association.

A prospective cohort study conducted at an academic infertility clinic in the United States [22] included 188 infertile women, 51 of whom had unexplained infertility (27%). All participants underwent serological screening. Among the 188 women with overall infertility, 3 (1.6%) tested positive for TGA. The expected prevalence of CeD in a similarly aged female population from the same geographical region was 1.3%. However, in a subgroup analysis of women with unexplained infertility, the prevalence rose to 3.9% (2 out of 51 patients).

In a cross-sectional study conducted in India, 230 women with unexplained infertility diagnosed according to WHO criteria 1993, considering normal semen analysis from the partner, and 305 control women were tested for seropositivity [23]. Elevated TGA levels were found in 13 women with unexplained infertility (5.7%) and 4 controls (1.3%), a statistically significant difference (*p* = 0.004). This indicated a 4.5-fold higher prevalence of CeD in infertile women compared to controls. Interestingly, none of the TGA-positive women exhibited classic gastrointestinal symptoms of CeD. The authors concluded that women with unexplained infertility might have subclinical CeD detectable through serological screening.

A Brazilian cross-sectional study of 170 infertile women, including 29 with unexplained infertility, reported a seropositive CeD prevalence of 2.9% (5 out of 170) in the overall cohort and 10.3% (3 out of 29) in the unexplained infertility group [24]. The authors suggested that these findings support the implementation of serological screening for CeD in women with unexplained infertility.

Similarly, a case-control study in Mexico evaluated 171 women with fertility disorders and 171 fertile control women matched by age for CeD-specific antibodies [25]. The results showed that seven infertile patients (4.1%) and only one control woman (0.6%) tested positive for elevated TGA levels.

Two studies from Iran provided additional insights into the prevalence of CeD among infertile women. In one study, serum samples from 100 couples with unexplained infertility and 200 fertile control couples were tested for TGA antibodies [19]. Positive TGA was detected in seven infertile women (7%) and seven control women (3.5%). Another prospective study involving 100 couples with unexplained infertility found that eight women (8%) tested positive for TGA, a prevalence higher than that observed in the general population [26]. Based on these findings, the authors recommended screening all women with unexplained infertility for CeD.

In summary, the results on the prevalence of CeD in women with overall or unexplained infertility revealed that 43 out of 959 (4.5%) had a positive TGA serology (six studies) in comparison to 676 control cases with 12 seropositive women (1.8%) (three studies). In case of unexplained infertility, 30 out of 481 subjects (6.2%) had a positive TGA serology. These findings indicated a significantly increased risk of seropositive CeD in women with infertility and advised that patients with unexplained infertility, particularly, should be screened for CeD.

Another Iranian study, where 8/100 females with infertility tested positive for CeD serology, showed a significant association between high levels of Anti-TTG Ab and infertility (odds ratio = 17.30, 95% CI: 2.13–140.39) [27].

A cross-sectional study conducted in China [28], which collected clinical and biochemical data from a total of 67 females with CeD and 67 healthy patients, showed significantly lower levels of anti-Müllerian hormone and higher Prolactin levels in the CeD group (all *p* < 0.05).

A nationwide Danish matched cohort study [29] involving 6319 women diagnosed with CeD and 63166 controls showed that prior to being diagnosed, CeD women had an excess risk of spontaneous abortion and extra stillbirths per 1000 pregnancies compared with the non-CeD women, indicating that that undiagnosed CeD can affect female reproduction.

The meta-analysis conducted by Singh et al. [30], which included nine studies published up to 2014, found that the prevalence of CeD in women with overall infertility (n = 884) was 2.3%, while the prevalence in women with unexplained infertility (n = 623) was 3.2%. Women with overall infertility were 3.5 times more likely, and those with unexplained infertility were 6 times more likely, to have CeD compared to fertile controls.

Another meta-analysis by Castano et al. [6] reviewed data from 23 scientific articles published up to 2019. The analysis reported similar prevalence rates of CeD-specific serology in women with overall infertility (n = 5935) and those with unexplained infertility (n = 1982), with pooled proportions ranging from approximately 1.3% to 1.6%. This represented a 3-fold increase in the odds of CeD in infertile women compared to the control population (n = 7202). Collectively, the findings of both meta-analyses confirm a higher risk of CeD in infertile women, particularly in those with unexplained infertility.

Conversely, two studies have demonstrated a higher prevalence of infertility in women with untreated CeD. Fortunato et al. [31] evaluated the fertility risk in 4070 women with CeD living in two regions of Southern Italy using ad hoc data and statistics that were routinely collected. They compared the frequency of hospital admissions for fertility issues in these women with 19,765 matched controls without CeD residing in the same region. The proportion of women hospitalized for fertility-related problems was significantly higher among those with CeD compared to age-matched controls (1.2% vs. 0.2%; *p* < 0.01).

In a more recent case-control study conducted in India between 2020 and 2021, Prasad et al. [32] used a detailed questionnaire to evaluate reproductive functions in 288 female patients with biopsy-confirmed CeD. They also included 586 age-matched healthy female controls. The study revealed that women with CeD had a significantly higher rate of current infertility compared to controls (10.5% vs. 5.2%; *p* = 0.028).

Lastly, a retrospective cohort study based on an online survey found higher rates of spontaneous abortion (50.6% vs. 40.6%; *p* = 0.01) and premature delivery (23.6% vs. 15.9%; *p* = 0.02) in women with celiac disease compared to controls [33].

#### 3.1.2. Negative Associations

The following six studies did not find a greater likelihood of CeD in women with infertility compared to controls or the general population.

In a large prospective cohort study conducted in the United States, 995 women undergoing in vitro fertilization were screened for CeD using a specific questionnaire and serological testing [34]. Among the participants, 24 patients (2.4%) tested positive for TGA. This prevalence was not significantly different from that observed in the general population (2.8%).

Another study enrolled 685 women at a Canadian fertility clinic to investigate the prevalence of CeD in women with unexplained infertility versus those with an identifiable cause of infertility [35]. Women on a GFD or with a previous CeD diagnosis were excluded. Eight of the 685 infertile women (1.2%) were identified as seropositive for CeD, including 4 out of 326 women with unexplained infertility (1.2%) and 4 out of 359 women with a known cause of infertility (1.1%). The study concluded that CeD was no more common in women with unexplained infertility than in those with a known infertility cause.

A study conducted in Iran examined 150 women with unexplained infertility and 150 control women for TGA serum levels [36]. Seropositive CeD was identified in three infertile patients (2.0%) and none of the controls, but the difference was not statistically significant (*p* = 0.49).

Three additional studies of infertile couples visiting fertility clinics provided further insight into the frequency of CeD in women. Hogen Esch et al. [37] screened 1038 subfertile male–female couples in the Netherlands for IgA TGA and IgA EMA. The prevalence of seropositive CeD among the women was 0.6% (6 out of 1038), which was not significantly higher than in the general Dutch population.

In a prospective pilot study, 65 Turkish couples with unexplained infertility were examined for CeD-specific antibodies, including TGA [38]. None of the women tested positive for any of the CeD-specific antibodies.

Finally, a cross-sectional study in Denmark estimated the prevalence of unrecognized CeD among infertile couples using TGA testing [39]. Among the 455 participating women, 4 (0.9%) were antibody-positive, comparable to the Danish general population.

In summary, these six studies revealed that among 3388 women with overall or unexplained infertility, 44 subjects (1.3%) had a positive TGA serology, which appears to be not higher than that of the general population. This finding did not support the routine screening of women with infertility for CeD. Only one of these studies included a control group, and none of the participating 150 subjects was seropositive [36].

A recent systematic review with meta-analysis of the prevalence of CeD in women with infertility also demonstrated that the prevalence of CeD in infertile women was not increased compared with controls [40]. Overall, 20 studies, published until February 2020, were included in the analysis. Based on 11 studies (1617 women), the pooled prevalence of biopsy-confirmed CeD was 0.7% in women with overall infertility. Restricting the study population to women with unexplained infertility, the pooled prevalence of biopsy-confirmed CeD was 0.6%. Regarding studies where CeD had been defined per serology (20 studies; 5158 women), the pooled prevalence of CeD was 1.1% in women with any infertility. In conclusion, results indicated that the prevalence of CeD was not higher in infertile women compared with the general population.

Vice versa, two investigations examined the prevalence of infertility in women with CeD. A large population-based study from the United Kingdom included 2,426,225 women of child-bearing age; 6506 women (0.3%) had a diagnosis of CeD [41]. The results revealed that the rate of infertility among women with CeD (4.4%) was similar to that of women without CeD (4.1%). As an exception, the rate was significantly higher among women diagnosed with CeD when they were 25–29 years old compared with women in the same age group without CeD. A Slovenian retrospective case-control study, including 205 women with biopsy-confirmed CeD and 103 healthy women, demonstrated that 23 subjects with CeD (11.2%) reported being treated for infertility, which was not statistically significant (*p* = 0.425) when compared with 8 control subjects (7.8%) [42].

A recent Italian survey of 493 celiac women revealed links between untreated celiac disease and miscarriages, anemia, low birth weight, and infertility; 73% felt poorly informed by healthcare professionals about reproductive risks [43].

Nahan et al. [44] identified >25,000,000 outpatient women without CeD, and 9368 with CeD from the database of 80 healthcare organisations. They reported that women with CeD had higher odds of later women’s health conditions including absent/rare menstruation (4.6% vs. 2.0%; OR 2.34), infertility (1.4% vs. 0.9%; OR 1.69), polycystic ovarian syndrome (3.3% vs. 1.0%; OR 3.2), menopausal disorders (4.3% vs. 1.56%; OR 285), and primary ovarian failure (0.96% vs. 0.16%; OR 6.25).

Mendelian randomization analysis, including data from 12,041 CD cases and 12,228 controls, from 14,759 infertile women, and from 111,583 controls, obtained from FinnGen and UK Biobank databases, revealed that there was no discernible link between genetic susceptibility to CD and female infertility [45]. The authors concluded that this finding implied a restrained basis for advocating CeD screening among women facing infertility issues.

Several studies and meta-analyses demonstrated a trend towards an increased prevalence of CeD in women with infertility and, vice versa, of infertility in women with active CeD, particularly in cases of unexplained infertility. Other investigations indicated that CeD is not more common in infertile women than in the general population. Figure 2 summarizes the results of a selection of studies about female infertility. All the papers published have some strengths and limitations. Screening for CeD in infertile women has been judged in a variety of different ways, and a widespread consensus is lacking. Further targeted prospective studies are needed to investigate the association between CeD and infertility in women.

**Table 1 nutrients-17-01575-t001:** Frequency of TGA-positive women with overall and unexplained infertility compared with fertile controls.

Author (Year) [Ref.]	Country ^a^	Overall Infertility		Unexplained Infertility		Controls	
		n ^b^	TGA (%)	n ^b^	TGA (%)	n ^b^	TGA (%)
Choi (2011) [22]	US	188	3 (1.6)	51	2 (3.9)	-	-
Hogen Esch (2011) [37]	NL	1038	6 (0.6)	-	-	-	-
Khoshbaten (2011) [19]	IR	-	-	100	7 (7.0)	200	7 (3.5)
Kumar (2011) [23]	IN	-	-	230	13 (5.7)	305	4 (1.3)
Machado (2013) [24]	BR	170	5 (2.9)	29	3 (10.3)	-	-
Karaca (2015) [38]	TR	-	-	65	0 (0.0)	-	-
Sabzevari (2017) [26]	IR	-	-	100	8 (8.0)	-	-
Grode (2018) [29]	DN	455	4 (0.9)	-	-	-	-
Gunn (2018) [35]	CA	685	8 (1.2)	326	4 (1.2)	-	-
Juneau (2018) [34]	US	995	24 (2.4)	-	-	-	-
Farzaneh (2019) [36]	IR	-	-	150	3 (2.0)	150	0 (0.0)
Remes-Troche (2023) [25]	MX	171	7 (4.1)	-	-	171	1 (0.6)

^a^ Country code ISO 3166-1. ^b^ Number of subjects. Legend: TGA: tissue transglutaminase antibodies; US: United States; NL: Netherlands, IR: Iran, IN: India, BR: Brazil, TR: Turkey, DN: Denmark, CA: Canada, MX: Mexico.

### 3.2. Infertility in Men

The underlying causes of infertility in men with CeD remain unclear and may involve multiple factors. A significant aspect of this increased infertility is represented by abnormalities in sperm parameters, including concentration, motility, and morphology. Beyond the general malabsorption observed in patients with active CeD, deficiencies in essential elements such as selenium and zinc are potential contributors to male infertility. Zinc, for instance, plays a critical role in various stages of sperm development, from germ cell maturation to the spermiation process [46]. Therefore, evaluating selenium and zinc levels in male patients with CeD who are experiencing infertility could be a valuable component of preconception counseling. Oxidative stress has been identified as another key factor in male infertility, contributing to lower sperm functionality. Figure 1 summarizes the results of a selection of studies about male infertility. A case-control study conducted in Iran investigated this phenomenon by comparing semen samples collected from 11 fertile men without CeD and 10 men affected by CeD [47]. The findings indicated that men with CeD exhibited significantly impaired sperm chromatin maturation, characterized by persistent histones and protamine deficiency, as well as increased DNA damage, compared to fertile controls (*p* < 0.05). Furthermore, the sperm viability percentage was significantly lower in men with CeD than in fertile controls (*p* < 0.05).

Studies regarding the fertility of men with CeD are relatively scarce. In 2011, Zugna et al. [48] assessed the fertility in a Swedish cohort of 7121 men with biopsy-confirmed CeD (Marsh type 3). The study group was compared with 31,677 men without CeD matched for age. The number of their children was the main outcome measure of the study. Results showed that men with CeD had a total of 9935 children, averaging 1.40 children per man, compared to 42,245 children, or 1.33 per man, in the control group. Furthermore, approximately 38% of men with CeD had no offspring, a proportion comparable to the reference population (39%). The study concluded that men with diagnosed CeD exhibit normal fertility rates.

Five studies that investigated the prevalence of CeD in couples with reproductive complications (see Section 3.3) provided information about infertility of male participants with CeD. In one of these studies, serum samples were analyzed from 100 Iranian couples with unexplained infertility [19]. As a comparison, 200 couples with no reported reproductive issues and at least one uncomplicated birth served as the control group. TGA were detected in six men (6%) from the study group and four men (4%) from the control group.

In another Iranian study, also including 100 couples with unexplained infertility, participants were tested for TGA serum levels [26]. In patients with positive serologic tests, a duodenal biopsy was performed to confirm the diagnosis. Overall, six men (6%) had positive TGA, and CeD was verified by duodenal biopsy in four subjects.

While both studies revealed a higher prevalence of CeD compared to the general population, neither was statistically significant.

In contrast, the other three studies did not find an association between infertility and CeD. Hogen Esch et al. [37] determined the prevalence of unrecognized CeD in 1038 subfertile male–female couples by testing IgA TGA and IgA EMA levels. Four men (0.4%) tested positive, a rate which was not higher than that of the adult population of the Netherlands. A total of 65 Turkish couples with unexplained infertility were tested for TGA and other CeD-specific antibodies [38]. Only one male partner (1.5%) was seropositive, and histological findings were compatible with Marsh type 3a. Screening for CeD in Danish couples with fertility problems, using TGA testing, revealed that 4 out of 438 men (0.9%) were antibody-positive [39]. CeD was confirmed by biopsy in three men. The prevalence of unrecognized CeD was equivalent to that of the Danish adult population. Altogether, further comprehensive studies should clarify the association between male infertility and CeD.

### 3.3. Infertility in Couples

A large cohort study conducted in the Netherlands investigated the prevalence of undiagnosed CeD in 1038 subfertile male–female couples attending a fertility clinic. Screening was performed using IgA TGA and IgA EMA [37]. The overall prevalence of CeD in this cohort was 0.48% (six females and four males), which was not significantly higher than the general adult population prevalence in the Netherlands (0.35%). Additionally, no significant association was found between CeD and subfertility between genders.

Another study, conducted in Iran, assessed serum samples from 100 couples with unexplained infertility to evaluate the prevalence of CeD [19]. The control group consisted of 200 couples without reproductive complications who had delivered at least one uncomplicated birth. Positive results of TGA were detected in 13 individuals from the infertile group (6.5%) and in 11 controls (2.8%), with a statistically significant difference (*p* = 0.027). The odds ratio (OR) for CeD in couples with unexplained infertility was 2.39, indicating a significantly higher seroprevalence of CeD in the infertile group compared to the fertile controls.

In a study involving 65 Turkish couples with unexplained infertility who were admitted to a fertility clinic, participants were tested for TGA and other CeD-specific antibodies [38]. Endoscopy was performed in patients with positive serology. Five couples had at least one positive antibody test; however, no couple showed antibody positivity in both partners simultaneously. Among the male participants, only one tested positive for all the antibodies. Histopathological examination of the antibody-positive patients revealed that only this male participant showed findings consistent with Marsh type 3a. Ultimately, a diagnosis of CeD was confirmed in only one couple (1.5%), with no female patients being diagnosed with CeD.

A prospective cross-sectional study from Iran included 100 couples with unexplained infertility who were tested for TGA levels [26]. In patients with positive serologic tests, a duodenal biopsy was performed to confirm the diagnosis. Overall, 14 patients (7%) with unexplained infertility had positive TGA (8 women and 6 men). In 11 out of 200 patients (5.5%), endoscopic finding was compatible with biopsy-verified CeD (7 women, 4 men).

A cross-sectional study was conducted to estimate the prevalence of unrecognized CeD among Danish couples referred for fertility treatment. Screening involved TGA testing, followed by small-bowel biopsies to confirm the diagnosis [39]. Out of 893 participants (51% women), 4 men and 4 women (0.90%) tested positive for TGA. Small-bowel biopsies were performed on seven seropositive individuals, corresponding to an overall prevalence of 0.45%. The prevalence of unrecognized CeD was equivalent to that of the Danish general population (0.48%).

Overall, two studies showed a positive association between infertility of couples and CeD: Around 7% of 200 couples reported problems with infertility [19,26]. In contrast, three studies did not find any significant relation. Only 0.5–1.5% of couples with reproductive problems were identified to have CeD, which corresponded to the general population [37,38,39]. Figure 2 summarizes the results of a selection of studies about couple infertility. Further investigations to address this issue are required.

## 4. Pathophysiological Mechanism

The etiology of infertility in CeD is multifactorial and remains only partially understood. Two main mechanisms have been proposed: micronutrient deficiencies and immune–endocrine dysregulation.

Micronutrient malabsorption is common in untreated CeD and includes iron, folate, selenium, zinc, and fat-soluble vitamins. These elements are crucial for gametogenesis, ovulation, implantation, and fetal development [49]. Zinc and selenium, in particular, influence sperm maturation and endometrial receptivity.

Immune-mediated and endocrine factors may further compromise reproductive health. Autoantibodies such as TGAs and anti-thyroid antibodies, as well as increased pro-inflammatory cytokines (e.g., IL-6, TNF-α), can impair endometrial angiogenesis and placentation. Hormonal disturbances, including altered prolactin, FSH, and LH levels, have also been observed [50,51].

Moreover, CeD is associated with menstrual irregularities, delayed menarche, and early menopause, possibly narrowing the reproductive window [52]. In men, oxidative stress and sexual dysfunction—including erectile and ejaculation disorders—may also contribute to reduced fertility.

Sexual dysfunction in women, including decreased sexual desire, dyspareunia (pain during intercourse), and anorgasmia (absence of orgasm), may contribute to infertility in patients with CeD [53]. Additionally, untreated CeD has been associated with delayed menarche and premature menopause, leading to a shorter fertile window compared to women without CeD [54]. In men, increased oxidative stress—a recurring finding in infertile males—has been linked to reduced sperm quality and function. Male infertility factors in CeD may also include erectile dysfunction and impaired sperm parameters, such as reduced quality and motility. Despite these observations, understanding the pathophysiological mechanisms underlying infertility in CeD remains limited, highlighting the need for further research.

## 5. Effect of a GFD

Morris et al. [18] were the first to report a link between untreated CeD and female infertility, as well as the positive impact of a GFD. They described three cases of infertile women with untreated CeD who conceived after adopting a GFD. Subsequent studies have demonstrated the beneficial effects of a GFD in improving fertility in women with CeD [7]. For instance, Nenna et al. reported four women (aged 28–39 years) in Italy who sought infertility treatment for periods ranging from 2 to 12 years and were subsequently diagnosed with CeD [55]. Remarkably, all four women conceived after adhering to a GFD for 2–9 months. One notable case involved a 39-year-old woman who had unsuccessfully tried to conceive for 11 years, including 4 years undergoing in vitro fertilization who, after following a GFD for 2 years, successfully delivered a baby.

A prospective cohort study conducted at an academic infertility clinic in the USA included 188 infertile women who underwent serologic screening for CeD [20]. Patients with positive serologic tests were advised to confirm the diagnosis with small-intestinal biopsies. Four women were diagnosed with CeD, confirmed by biopsy, and received nutritional counseling to adopt a GFD. Remarkably, all four women conceived within a year following their diagnosis and dietary changes.

In a retrospective study from a Spanish infertility clinic, the impact of a GFD was assessed in women with CeD who had experienced recurrent implantation failure [56]. Data from women following a gluten-containing diet (GCD) were compared with those adhering to a GFD. Significant differences were observed: the live birth rate was 0% (0/19) for women on a GCD versus 60% (6/10) for those on a GFD (*p* = 0.004), and the miscarriage rate was 100% (3/3) for the GCD group compared to 14% (1/7) in the GFD group (*p* = 0.033).

A retrospective study from Morocco demonstrated that reproductive disorders in patients with celiac disease are frequent (58/173 patients) but largely reversible, with 90% of cases (26 patients out of 29) showing improvement following the initiation of a gluten-free diet [57].

However, contrasting results emerged from a USA study of 28 women with seropositive CeD undergoing in vitro fertilization [34]. Outcomes in women adhering to a GFD (n = 3) were comparable to those on a GCD. Parameters such as mature oocytes retrieved (10.2 vs. 10.8), fertilization rates (82.4% vs. 83.6%), and blastulation rates (54.8% vs. 48.6%) showed no significant differences (*p* = 0.37–0.58). A notable limitation of this study, and others, was the lack of rigorous testing for strict GFD adherence.

In conclusion, further investigations involving larger cohorts of CeD patients, including those strictly adherent and non-adherent to a GFD, are required to clarify the role of a GFD in addressing fertility issues in women with CeD.

## 6. Discussion

Despite decades of research, the association between CeD and infertility remains controversial. Some studies show increased prevalence of CeD among women with unexplained infertility, while others report no difference from the general population. These discrepancies may stem from variability in diagnostic criteria, population characteristics, and methodology.

A limitation of our review is the use only of PubMed and Scopus as a source database, which may have limited the comprehensiveness of the literature search. Furthermore, as a narrative review, study selection and data interpretation are inherently subjective, introducing potential bias. Other limitations in the research include small sample sizes and the absence of control groups.

The prevalence of CeD varies geographically and ethnically, ranging from under 0.1% in East Asia to over 1% in Europe and the Middle East. The low prevalence of CeD in non-Caucasian women may reflect a combination of genetic, diagnostic, and healthcare disparities rather than biological differences. While CeD is more common in white populations due to genetic predisposition [58], underdiagnosis due to limited access to testing in non-Caucasian populations may be exaggerating the disparity. Non-Caucasian women with CeD often depend on community health centers rather than primary care settings, where access to diagnostic tools may be limited. As a result, being female, non-Caucasian, and having undiagnosed celiac disease likely creates a triple burden, increasing the risk of delayed diagnosis and inadequate care [59].

These disparities must be taken into account when evaluating the clinical utility of screening for CeD in infertile populations. A “one-size-fits-all” approach is not appropriate, and pretest probability should guide testing. Moreover, many studies rely primarily on serological testing without confirming diagnoses through duodenal biopsy. The standard laboratory tests for CeD screening have commonly included IgA TGA, IgA EMA, and total IgA antibodies. In recent years, rapid point-of-care (POC) tests have been proposed as alternatives to standard testing. However, Grode et al. evaluated the diagnostic accuracy of the POC test (Simtomax^®^) and found it unsuitable due to a high rate of false-positive results (sensitivity = 42.9%) [60].

Another confounding factor is that some studies failed to clearly distinguish between infertility and sexual dysfunction. CeD may not inherently reduce reproductive capacity but associated depression, fatigue, or dyspareunia could affect sexual activity, thereby reducing conception rates.

While routine screening for CeD in infertile patients remains controversial, several studies have demonstrated the positive impact of a GFD on fertility in women with CeD. Adhering to a strict GFD has been shown to improve the chances of conception and the birth of a healthy child. Patients must also be informed of the importance of dietary compliance, as any lapses in following a GFD can significantly reduce their likelihood of achieving a successful outcome [61].

In summary, further studies are required to evaluate the risk of unrecognized CeD in infertile patients to define the role of routine serological screening for CeD in infertile patients and to elucidate the underlying mechanism for infertility in active CeD. Future research should employ standardized infertility definitions; include control groups matched for age, region, and socio-economic status; and use structured protocols (e.g., PRISMA-ScR) with validated tools for bias assessment. Prospective studies that stratify patients by their adherence to a gluten-free diet and by reproductive outcomes are also warranted. Future research should consider the following points: the diagnosis of CeD should include serological standardized testing of TGA, EMA, and total IgA antibodies followed by confirmatory duodenal biopsy (Marsh criteria) in all seropositive patients. The definition of infertility should be based on the WHO definition or a close version. The sample size of participants should be more than 200 infertile patients and 200 controls with proven fertility, the latter being matched for age, socio-economic status, and regional origin.

## Figures and Tables

**Figure 1 nutrients-17-01575-f001:**
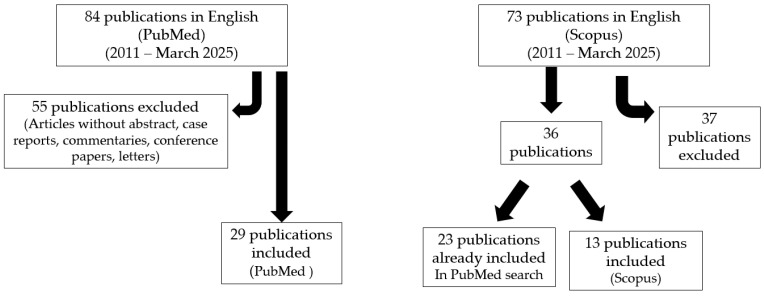
PRISMA flow diagram of search procedure.

**Figure 2 nutrients-17-01575-f002:**
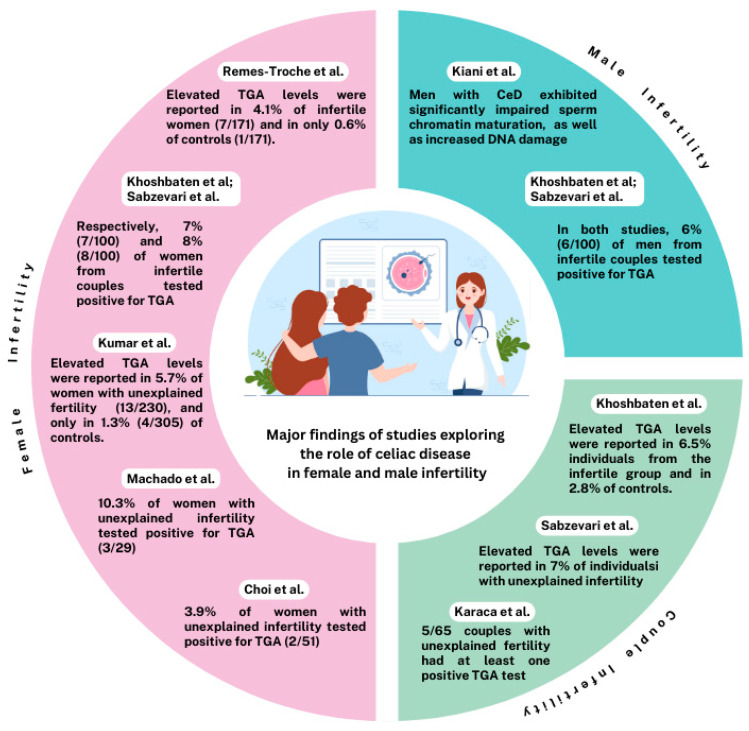
A selection of studies about CeD and female, male, and couple infertility.

## Data Availability

The original contributions presented in the study are included in the article; further inquiries can be directed to the corresponding author.

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
