# Peer review of "Fertility in Celiac Disease: The Impact of Gluten on Male and Female Reproductive Health"

_nutrients, 2025, doi:10.3390/nu17091575_

Round 1

Reviewer 1 Report

Comments and Suggestions for Authors
  1. Positioning of the Study among Existing Research
    This study is an ambitious attempt to gain a comprehensive insight into the potential relationship between celiac disease (CeD) and infertility. However, it seems unclear how this work positions itself in relation to the many previous studies on the same topic. It would be helpful if the authors could clearly highlight how their research differs from or expands upon existing literature, as this will clarify the value of publishing this study.

  2. Balance of Evidence in the Abstract
    Large prospective cohort studies, systematic reviews, and meta-analyses have reported that the prevalence of CeD in women with infertility is not necessarily higher than in control groups. The authors also cite and acknowledge these findings in the text. Nevertheless, the abstract appears to emphasize that the prevalence of infertility among CeD patients is significantly higher than in the general population, which may give the impression of a selective or unbalanced presentation of results. To avoid misunderstanding, a more neutral and cautious tone may be needed, reflecting the overall body of current research.

  3. Methodological Approach
    The study appears to rely on a traditional narrative review approach in which data extraction is left to the authors’ discretion. While such a strategy can be useful for a rapid overview, the validity of a purely narrative method may be questioned in an area like CeD and infertility, where a strong consensus has not been reached. Employing a more structured method—such as following a protocol with a transparent search strategy, standardized data extraction forms, and an assessment of risk of bias—could strengthen the clarity and importance of this review. The authors may wish to consider guidelines for scoping reviews (e.g., PRISMA-ScR) to enhance methodological rigor.

  4. Scope of the Title vs. Content
    In addition to autoimmune CeD, gluten-related disorders include wheat-dependent exercise-induced anaphylaxis (an allergic condition) and non-celiac gluten sensitivity (NCGS), in which neither allergy nor autoimmunity plays a central role. Because the present manuscript focuses mostly on CeD, the current title mentioning “gluten” and fertility might lead some readers to assume a broader scope than is actually covered. Please clarify whether the study aims to address gluten-related disorders in general or is limited to CeD, and adjust the title and content accordingly.

  5. Selection Bias in the Literature Search
    In the Materials and Methods, the authors describe identifying 80 initial papers, extracting 26 of them, and finally adding 13 more. However, there is no explanation of how selection bias was minimized during this process. It would be very helpful to provide more detail—preferably with a flowchart—on the inclusion and exclusion criteria so that readers can better understand how the final set of studies was chosen.

  6. Geographic and Ethnic Differences in CeD Prevalence
    It is well known that the prevalence of CeD varies greatly depending on geographic region and ethnicity. Therefore, even if testing for CeD in unexplained infertility might be beneficial in regions where CeD is relatively common, this may not apply to areas where it is extremely rare. Further discussion of regional differences and the role of pre-test probability would make the manuscript more comprehensive.

  7. Definition of Unexplained Infertility
    In the Results section, the authors briefly summarize the definition of infertility in the first paragraph, and then discuss the causes of infertility in the second paragraph, categorizing them into female and male factors. However, the paper does not clearly define unexplained infertility. In reproductive medicine, infertility is typically labeled “unexplained” when no identifiable factors are found in either partner after evaluating female ovulatory/tubal factors and male spermatogenic factors. The paper sometimes seems to imply that unexplained infertility pertains only to women. Please verify whether the original sources treat unexplained infertility as exclusively female or consider both partners, and integrate those findings accurately.

  8. Distinguishing Sexual Dysfunction from Infertility
    Various symptoms of CeD—such as abdominal pain, skin manifestations, reduced motivation, and depression—could affect a couple’s sexual life. Even if CeD does not inherently reduce reproductive potential, a decrease in sexual frequency could prolong the time to natural conception. Hence, if one wishes to demonstrate that CeD genuinely lowers fertility, assessing treatment success rates of intrauterine insemination or assisted reproductive technology might be more appropriate. Alternatively, if the claim is that CeD exerts adverse effects on sexual life and thereby indirectly reduces pregnancy rates, it would be important to distinguish clearly between infertility per se and sexual dysfunction (e.g., diminished libido, dyspareunia in women; diminished libido, erectile dysfunction, or ejaculation disorders in men).

  9. Pathophysiological Mechanisms
    The “Pathophysiological mechanism” section appears relatively brief compared to other parts of the manuscript. We suggest dividing the discussion into mechanisms related to micronutrient malabsorption in CeD versus other immunological and endocrine pathways, and describing the latest basic science insights on how each may contribute to infertility. Expanding this section would provide a more comprehensive understanding of the pathogenesis involved.

Author Response

Reviewer 1

  1. Positioning of the Study among Existing Research
    This study is an ambitious attempt to gain a comprehensive insight into the potential relationship between celiac disease (CeD) and infertility. However, it seems unclear how this work positions itself in relation to the many previous studies on the same topic. It would be helpful if the authors could clearly highlight how their research differs from or expands upon existing literature, as this will clarify the value of publishing this study.

We thank the reviewer for this important observation. In the revised version, we have clarified in the Introduction how our study differs from previous reviews. Specifically, we offer a broader perspective by including studies published up to March 2025 and by separately analyzing findings related to female, male, and couple infertility. Additionally, we highlight the inconsistencies and conflicting results within the literature, underscoring the need for a critical narrative overview rather than a quantitative synthesis.

  1. Balance of Evidence in the Abstract
    Large prospective cohort studies, systematic reviews, and meta-analyses have reported that the prevalence of CeD in women with infertility is not necessarily higher than in control groups. The authors also cite and acknowledge these findings in the text. Nevertheless, the abstract appears to emphasize that the prevalence of infertility among CeD patients is significantly higher than in the general population, which may give the impression of a selective or unbalanced presentation of results. To avoid misunderstanding, a more neutral and cautious tone may be needed, reflecting the overall body of current research.

We agree with the reviewer and have revised the Abstract to adopt a more cautious and neutral tone. The updated version acknowledges the conflicting evidence in the literature and emphasizes that while some studies show a higher prevalence of unexplained infertility in CeD patients, others report no significant differences from controls.

  1. Methodological Approach
    The study appears to rely on a traditional narrative review approach in which data extraction is left to the authors’ discretion. While such a strategy can be useful for a rapid overview, the validity of a purely narrative method may be questioned in an area like CeD and infertility, where a strong consensus has not been reached. Employing a more structured method—such as following a protocol with a transparent search strategy, standardized data extraction forms, and an assessment of risk of bias—could strengthen the clarity and importance of this review. The authors may wish to consider guidelines for scoping reviews (e.g., PRISMA-ScR) to enhance methodological rigor.

We have clarified in the Materials and Methods that this is a narrative review. We also acknowledge the limitations of this approach and explicitly recommend that future studies employ a structured scoping or systematic review method using PRISMA-ScR guidelines for enhanced methodological rigor. We added a Figure (Figure 1) reporting PRISMA flow diagram of search procedure.

  1. Scope of the Title vs. Content
    In addition to autoimmune CeD, gluten-related disorders include wheat-dependent exercise-induced anaphylaxis (an allergic condition) and non-celiac gluten sensitivity (NCGS), in which neither allergy nor autoimmunity plays a central role. Because the present manuscript focuses mostly on CeD, the current title mentioning “gluten” and fertility might lead some readers to assume a broader scope than is actually covered. Please clarify whether the study aims to address gluten-related disorders in general or is limited to CeD, and adjust the title and content accordingly.

We thank the reviewer for this remark. The title has been revised to: Fertility in Celiac Disease: The Impact of Gluten on Male and Female Reproductive Health" to more accurately reflect the manuscript’s focus on autoimmune CeD rather than the broader category of gluten-related disorders.

  1. Selection Bias in the Literature Search
    In the Materials and Methods, the authors describe identifying 80 initial papers, extracting 26 of them, and finally adding 13 more. However, there is no explanation of how selection bias was minimized during this process. It would be very helpful to provide more detail—preferably with a flowchart—on the inclusion and exclusion criteria so that readers can better understand how the final set of studies was chosen.

We have included a simplified PRISMA-style flowchart to illustrate the selection process. Furthermore, we now explicitly state the inclusion and exclusion criteria used, and we recognize that data extraction was based on authors’ discretion, which may introduce selection bias. These limitations are now discussed in the Discussion section.

  1. Geographic and Ethnic Differences in CeD Prevalence
    It is well known that the prevalence of CeD varies greatly depending on geographic region and ethnicity. Therefore, even if testing for CeD in unexplained infertility might be beneficial in regions where CeD is relatively common, this may not apply to areas where it is extremely rare. Further discussion of regional differences and the role of pre-test probability would make the manuscript more comprehensive.

We appreciate this valuable suggestion. We have added a specific paragraph in the Discussion that addresses how CeD prevalence differs by geography and ethnicity. We also discuss how pre-test probability should guide CeD screening decisions, particularly in the context of unexplained infertility.

  1. Definition of Unexplained Infertility
    In the Results section, the authors briefly summarize the definition of infertility in the first paragraph, and then discuss the causes of infertility in the second paragraph, categorizing them into female and male factors. However, the paper does not clearly define unexplained infertility. In reproductive medicine, infertility is typically labeled “unexplained” when no identifiable factors are found in either partner after evaluating female ovulatory/tubal factors and male spermatogenic factors. The paper sometimes seems to imply that unexplained infertility pertains only to women. Please verify whether the original sources treat unexplained infertility as exclusively female or consider both partners, and integrate those findings accurately.

Thank you for pointing this out. We have revised the Results section to clearly define unexplained infertility as per WHO and reproductive medicine guidelines, specifying that it involves a lack of identifiable causes in both partners after comprehensive evaluation. We also reviewed our citations to ensure alignment with this definition.

  1. Distinguishing Sexual Dysfunction from Infertility
    Various symptoms of CeD—such as abdominal pain, skin manifestations, reduced motivation, and depression—could affect a couple’s sexual life. Even if CeD does not inherently reduce reproductive potential, a decrease in sexual frequency could prolong the time to natural conception. Hence, if one wishes to demonstrate that CeD genuinely lowers fertility, assessing treatment success rates of intrauterine insemination or assisted reproductive technology might be more appropriate. Alternatively, if the claim is that CeD exerts adverse effects on sexual life and thereby indirectly reduces pregnancy rates, it would be important to distinguish clearly between infertility per se and sexual dysfunction (e.g., diminished libido, dyspareunia in women; diminished libido, erectile dysfunction, or ejaculation disorders in men).

We agree, and the Pathophysiological Mechanisms section has been expanded to better distinguish between biological infertility and potential sexual dysfunction resulting from CeD symptoms (e.g., depression, dyspareunia, erectile dysfunction). We also comment on the limitations of current data in isolating these effects.

  1. Pathophysiological Mechanisms
    The “Pathophysiological mechanism” section appears relatively brief compared to other parts of the manuscript. We suggest dividing the discussion into mechanisms related to micronutrient malabsorption in CeD versus other immunological and endocrine pathways, and describing the latest basic science insights on how each may contribute to infertility. Expanding this section would provide a more comprehensive understanding of the pathogenesis involved.

The Pathophysiological Mechanism section has been substantially revised. It now separates mechanisms into: micronutrient deficiencies (e.g., zinc, folate, selenium), and immune and endocrine dysregulation (e.g., cytokines, hormonal imbalances, autoantibodies).
We have also added recent evidence on endometrial angiogenesis and sperm oxidative stress, enhancing the depth of discussion.

Reviewer 2 Report

Comments and Suggestions for Authors

The review article titled “nutrients-3572360_Celiac disease and reproductive health: exploring the impact of gluten on fertility” is submitted to the “Clinical Nutrition” of the Special Issue “Advances in Prevention and Management of Celiac Disease”.

The primary objective of this review is to synthesize recent studies exploring the relationship between coeliac disease (CeD) and infertility, with the aim of increasing physicians' awareness regarding the potential presence of CeD in patients experiencing reproductive difficulties.

The abstract is a crucial section for disseminating the work, as it remains freely accessible across all platforms. Therefore, it is advisable that the abstract provides a more comprehensive preview of the study's content, including background information, objectives, the type of review conducted, the databases utilized, the time frame covered, key findings, and conclusions. Accordingly, the abstract should be restructured to present a more thorough overview of the study.

The introduction successfully highlights the significance of the topic and the hypothesis underpinning the study, while also addressing discrepancies and existing literature through appropriate bibliographic references.

In the materials and methods section, the review design is not explicitly stated. It appears to be a narrative review, but this should be confirmed by the authors. The specific type of review conducted should be clearly indicated.

The reliance on a single database represents a significant limitation of this study. Current recommendations for systematic reviews suggest the use of at least three databases. Additionally, the inclusion and exclusion criteria should be explicitly stated.

The results section contains concepts that would be more appropriately placed in the introduction. The results section should strictly present the findings obtained from the review.

Regarding the tables, all abbreviations should be explained in footnotes to ensure they are self-explanatory. Furthermore, the extensive results should be summarized with the aid of tables to facilitate synthesis and clarity.

The discussion section is limited and should include a reflection on the methodological constraints, particularly the reliance on a single database. Additionally, the discussion should incorporate relevant literature proposing explanatory hypotheses regarding the potential coherence of the relationship between CeD and infertility. It would also be beneficial to outline possible future research strategies on this topic. The discussion should avoid recommending the use of diagnostic tests.

Overall, this is a highly relevant topic; however, the study is significantly constrained by its dependence on a single database in this type of review.

Comments on the Quality of English Language

I have no comments on the quality of the English, but I do not consider myself an expert in the language.

Author Response

Reviewer 2

The review article titled “nutrients-3572360_Celiac disease and reproductive health: exploring the impact of gluten on fertility” is submitted to the “Clinical Nutrition” of the Special Issue “Advances in Prevention and Management of Celiac Disease”.

The primary objective of this review is to synthesize recent studies exploring the relationship between coeliac disease (CeD) and infertility, with the aim of increasing physicians' awareness regarding the potential presence of CeD in patients experiencing reproductive difficulties.

The abstract is a crucial section for disseminating the work, as it remains freely accessible across all platforms. Therefore, it is advisable that the abstract provides a more comprehensive preview of the study's content, including background information, objectives, the type of review conducted, the databases utilized, the time frame covered, key findings, and conclusions. Accordingly, the abstract should be restructured to present a more thorough overview of the study.

We thank the reviewer for his/her suggestions. We have restructured the Abstract to explicitly include background context, study aim, review type (narrative review), database used (PubMed), time frame (2011–2025), main findings, and conclusions. The tone has been balanced to reflect both positive and negative findings from the literature.

The introduction successfully highlights the significance of the topic and the hypothesis underpinning the study, while also addressing discrepancies and existing literature through appropriate bibliographic references. In the materials and methods section, the review design is not explicitly stated. It appears to be a narrative review, but this should be confirmed by the authors. The specific type of review conducted should be clearly indicated.

As suggested, we now explicitly state in the Materials and Methods section that this is a narrative review. We also offer a broader perspective by including studies published up to March 2025 and acknowledge this design’s limitations, suggesting the use of structured methods (e.g., PRISMA-ScR) for future research.

The reliance on a single database represents a significant limitation of this study. Current recommendations for systematic reviews suggest the use of at least three databases. Additionally, the inclusion and exclusion criteria should be explicitly stated.

We acknowledge this limitation in both the Methods and Discussion sections. The choice of PubMed was due to resource constraints, and we recognize that additional databases such as Embase or Scopus may have yielded further relevant studies.

The results section contains concepts that would be more appropriately placed in the introduction. The results section should strictly present the findings obtained from the review.

We thank the reviewer for this observation. We have revised the Results section to strictly present findings, and we have relocated contextual/background information to the Introduction, where appropriate.

Regarding the tables, all abbreviations should be explained in footnotes to ensure they are self-explanatory. Furthermore, the extensive results should be summarized with the aid of tables to facilitate synthesis and clarity.

All tables now include footnotes clarifying abbreviations such as TGA, n, country codes, etc., to enhance reader understanding and ensure self-explanatory presentation.

The discussion section is limited and should include a reflection on the methodological constraints, particularly the reliance on a single database. Additionally, the discussion should incorporate relevant literature proposing explanatory hypotheses regarding the potential coherence of the relationship between CeD and infertility. It would also be beneficial to outline possible future research strategies on this topic. The discussion should avoid recommending the use of diagnostic tests. Overall, this is a highly relevant topic; however, the study is significantly constrained by its dependence on a single database in this type of review.

We have substantially expanded the Discussion section to:

- Reflect on the limitations of using a single database and a narrative approach.

- Discuss potential mechanisms (nutritional, immune, endocrine, sexual) based on the reviewed literature;

- Recommend directions for future studies including prospective design, broader databases, and structured protocols.

We have also avoided recommending widespread screening, in line with the reviewer’s guidance.

Reviewer 3 Report

Comments and Suggestions for Authors

This is an interesting review assessing Celiac disease and possible causes of infertility in men and women. It is understandable that in this immune mediated malabsorption syndrome, nutrient supplementation and special diets are essential for optimization of general health for optimal preconception health.

Some points for the authors to consider:

  1. Would the authors identify certain populations or ethnicities who are more predisposed to Celiac disease and this should be routinely screened?
  2. Would the authors suggest GFD routinely for certain types of fertility problems or populations when they are trying to conceive even if they are not diagnosed with celiac disease?
  3. Would the authors recommend a preconception diet suitable for couples with unexplained infertility and recommend a certain diagnostic approach for these couples or the existing data are too little to make clinically meaningful recommendations?

These points should be addressed in the manuscript and developed further in the discussion would enrich the discussion.

Author Response

REVIEWER 3

This is an interesting review assessing Celiac disease and possible causes of infertility in men and women. It is understandable that in this immune mediated malabsorption syndrome, nutrient supplementation and special diets are essential for optimization of general health for optimal preconception health.

Some points for the authors to consider:

  1. Would the authors identify certain populations or ethnicities who are more predisposed to Celiac disease and this should be routinely screened?

Yes, thank you for this suggestion. We have added a paragraph to the Discussion highlighting that CeD is more prevalent in certain populations, including those of Northern European and Middle Eastern descent. We note that screening may be more justified in these groups, especially when unexplained infertility is present.

  1. Would the authors suggest GFD routinely for certain types of fertility problems or populations when they are trying to conceive even if they are not diagnosed with celiac disease?

Currently, there is insufficient evidence to recommend a gluten-free diet (GFD) for individuals without a confirmed celiac disease (CeD) diagnosis. We emphasize this point in the revised Discussion, referencing current guidelines and suggesting caution until more robust data are available.

  1. Would the authors recommend a preconception diet suitable for couples with unexplained infertility and recommend a certain diagnostic approach for these couples or the existing data are too little to make clinically meaningful recommendations?

We agree this is an important area for future exploration. In the Discussion, we clarify that while a nutritionally adequate diet is beneficial in all cases, current data are insufficient to recommend a specific CeD-focused preconception diet for all cases of unexplained infertility. We highlight the need for further prospective studies to investigate this possibility and define clear diagnostic pathways.

Round 2

Reviewer 1 Report

Comments and Suggestions for Authors

I appreciate the effort you have put into addressing the concerns raised in the initial review. The new additions and clarifications appear to resolve the primary questions regarding the scope, methodology, and interpretation of results.

Author Response

We are grateful for your positive feedback and appreciation of our efforts to address the concerns raised in the initial round of review. We are pleased to know that the additions and clarifications we introduced have effectively resolved the primary questions regarding the scope, methodology, and interpretation of results.

Reviewer 2 Report

Comments and Suggestions for Authors

Thank you very much for allowing me to review the manuscript (nutrients-3572360) once again. The authors have made substantial improvements, and their considerable effort is clearly evident.

The revised title is now much more aligned with the content of the manuscript, which enhances its coherence.

However, there remain significant limitations—most notably, the fact that this is a narrative review relying solely on PubMed as the source of literature. This substantially restricts the methodological rigour and comprehensiveness expected for publication in a first-quartile journal.

At the very least, additional databases should have been consulted to strengthen the review's scope and reliability.

Author Response

Thank you very much for your thoughtful comments and for acknowledging the improvements made to the manuscript, particularly concerning its coherence and alignment between the title and content.

In accordance with your valuable suggestion, we have expanded our literature search to include Scopus in addition to PubMed. This enhancement has been incorporated into the Methods section and reflected in the updated Figure 1, which now presents the revised literature selection process. We believe this modification improves the methodological rigour and comprehensiveness of the review, addressing the limitation you identified.

Reviewer 3 Report

Comments and Suggestions for Authors

The authors have replied my concerns satisfactorily. 

Author Response

We sincerely appreciate your feedback and are glad that our revisions and responses have satisfactorily addressed your concerns.